# Relaxin-2 May Suppress Endometriosis by Reducing Fibrosis, Scar Formation, and Inflammation

**DOI:** 10.3390/biomedicines8110467

**Published:** 2020-10-31

**Authors:** Osamu Yoshino, Yosuke Ono, Masako Honda, Kyoko Hattori, Erina Sato, Takehiro Hiraoka, Masami Ito, Mutsumi Kobayashi, Kenta Arai, Hidekazu Katayama, Hiroyoshi Tsuchida, Kaori Yamada-Nomoto, Shunsuke Iwahata, Yoshiyuki Fukushi, Shinichiro Wada, Haruko Iwase, Kaori Koga, Yutaka Osuga, Michio Iwaoka, Nobuya Unno

**Affiliations:** 1Department of Obstetrics and Gynecology, Kitasato University School of Medicine, Kanagawa 252-0375, Japan; mhonda@kitasato-u.ac.jp (M.H.); kyoko17@kitasato-u.ac.jp (K.H.); erinast@med.kitasato-u.ac.jp (E.S.); thiraoka@med.kitasato-u.ac.jp (T.H.); iwahata@kitasato-u.ac.jp (S.I.); haiwase@med.kitasato-u.ac.jp (H.I.); unno@med.kitasato-u.ac.jp (N.U.); 2Department of Obstetrics and Gynecology, Teine Keijinkai Hospital, Hokkaido 006-0811, Japan; nadal.babolat@hotmail.co.jp (Y.O.); kohta294@yahoo.co.jp (Y.F.); wa_shin_2002@yahoo.co.jp (S.W.); 3Department of Obstetrics and Gynecology, University of Toyama, Toyama 930-0194, Japan; msmito@med.u-toyama.ac.jp (M.I.); k.mutsugoroo@gmail.com (M.K.); ultrahidehiro@yahoo.co.jp (H.T.); kaorinomoto319@gmail.com (K.Y.-N.); 4Department of Chemistry, School of Science, Tokai University, Tokyo 259-1292, Japan; ak697718@tsc.u-tokai.ac.jp (K.A.); miwaoka@keyaki.cc.u-tokai.ac.jp (M.I.); 5Department of Applied Biochemistry, Tokai University, Tokyo 259-1292, Japan; katay@tokai-u.jp; 6Department of Obstetrics and Gynecology, The University of Tokyo, Tokyo 113-8655, Japan; kawotan-tky@umin.ac.jp (K.K.); yutakaos-tky@umin.ac.jp (Y.O.)

**Keywords:** relaxin, endometriosis, fibrosis, p38MAPK, cAMP

## Abstract

Background: Relaxin (RLX)-2, produced by the corpus luteum and placenta, is known to be potentially effective in fibrotic diseases of the heart, lungs, kidneys, and bladder; however, its effectiveness in endometriosis has not yet been investigated. In the present study, we conducted a comprehensive study on the effect of RLX-2 on endometriosis. We checked the expressions of LGR-7, a primary receptor of RLX-2, in endometriomas using immunohistochemistry. Endometriotic stromal cells (ESCs) purified from surgical specimens were used in in vitro experiments. The effects of RLX-2 on ESCs were evaluated by quantitative-PCR, ELISA, and Western blotting. Gel contraction assay was used to assess the contraction suppressive effect of RLX-2. The effect of RLX-2 was also examined in the endometriosis mouse model. LGR-7 was expressed in endometriotic lesions. In ESCs, RLX-2 increased the production of cAMP and suppressed the secretion of interleukin-8, an inflammatory cytokine, by 15% and mRNA expression of fibrosis-related molecules, plasminogen activator inhibitor-1 (PAI-1), and collagen-I by approximately 50% (*p* < 0.05). In the gel contraction assay, RLX-2 significantly suppressed the contraction of ESCs, which was cancelled by removing RLX-2 from the medium or by adding H89, a Protein Kinase A (PKA) inhibitor. In ESCs stimulated with RLX-2, p38 MAPK phosphorylation was significantly suppressed. In the endometriosis mouse model, administration of RLX-2 significantly decreased the area of the endometriotic-like lesion with decreasing fibrotic component compared to non-treated control (*p* = 0.01). RLX-2 may contribute to the control of endometriotic lesion by suppressing fibrosis, scar formation, and inflammation.

## 1. Introduction

Endometriosis affects about 10% of women of reproductive age. Accumulating evidence suggests that immune cells, adhesion molecules, extracellular matrix metalloproteinase and pro-inflammatory cytokines activate or alter peritoneal microenvironment, creating the conditions for the development of ectopic endometrial cells [1]. Among the various symptoms caused by endometriosis, pain such as dysmenorrhea, dyspareunia, and dyschezia, and infertility are the significant symptoms of the disease, and these may significantly attenuate their quality of life (QOL) [2,3]. During the development and progression of endometriosis, collagen production is increased, further contributing to a stiff matrix and ultimately the formation of fibrosis [4]. These changes may occur as a secondary event triggered by inflammation due to the presence of ectopic endometrial cells in an affected tissue [5]. Interestingly, Lagana et al. reported that by analyzing macrophages, pro-inflammatory macrophages are dominant in the early stages of endometriosis, while pro-fibrotic macrophages are activated in the late stages [6].

Excess fibrosis appears as the phenomenon underpinning endometriosis-associated morbidity and some manifestations of the disease, such as pain and infertility through causing adhesion and scarring [7,8]. Moreover, fibrosis with concomitant loss of ovarian follicles is observed in ovaries with endometriomas [9], suggesting the alteration of ovarian function by the disease. In fact, women with endometrioma experience a progressive decline in serum anti-Müllerian hormone (AMH) levels, a marker of ovarian function, faster than that in healthy women [10]. Collectively, fibrosis is at the center of the pathophysiology of endometriosis and the suppression of the fibrosis formation could be a good target for the treatment in endometriosis. So far, there have been many candidates for the treatment of fibrosis, including the target of TGF-β due to the contribution of TGF-β in the pathogenesis of fibrosis. However, there is still a significant hurdle for the clinical transition of these candidates from the point of in vivo stability, tissue-specificity, and side effects [5].

Recently, relaxin (RLX)-2 has attracted significant interest for its anti-fibrotic effect in various fibrous diseases of the heart, lungs, kidneys, and bladder [11]. RLX-2 is a peptide hormone belonging to the insulin superfamily that has a collagenolytic effect [11,12]. In women, RLX-2 is secreted from the corpus luteum and placenta, and the role of RLX-2 during labor has been studied extensively. During pregnancy, the concentration of serum RLX-2 is increased substantially to a level of ng/mL [13]. In pregnant women, RLX-2 secreted by the placenta softens the ligaments of the pelvic joints, which allows the symphysis pubis to expand as much as 20 mm during labor, leading to easier delivery [12]. To date, few studies on RLX-2 in endometriosis have been reported, where the RLX-2 receptor, LGR-7, is expressed in endometriotic lesions [14]. However, the effect of RLX-2 on endometriosis remained unclear. Since RLX-2 has been shown to be a potential treatment for various fibrotic diseases [11], we conducted this study on the hypothesis that RLX-2 could be an effective treatment for endometriosis.

## 2. Material and Methods

The experimental procedures were approved by the institutional review board of Kitasato University (Sagamihara, Kanagawa, Japan, approved number: B18-265), and University of Toyama (Toyama, Toyama, Japan, approved number: 25-44) and signed informed consent for the use of samples was obtained from each patient. Animal experiments were approved by the ethical committee of Kitasato University (approved number: 1114, 2015-022). All experiments were performed in accordance with relevant guidelines and regulations. Endometrioma tissues were obtained from patients undergoing laparoscopic surgery. All women had regular menstrual cycles, and none had taken hormonal medications for at least three months before laparoscopy.

### 2.1. Reagents and Materials

Dulbecco’s Modified Eagle Medium (DMEM)/F12 was purchased from Sigma-Aldrich (St. Louis, MO, USA). Fetal bovine serum (FBS) was purchased from Life Technologies (Minato-ku, Tokyo, Japan). Antibiotics (a mixture of penicillin, streptomycin, and amphotericin B) were purchased from Wako Pure Chemical Industries (Chuo-ku, Osaka, Japan). Human recombinant RLX-2 (vehicle; 0.1% BSA/PBS) was purchased from R & D System (Minneapolis, MN, USA). RLX-2 for mouse experiment was chemically synthesized by the authors (Kenta Arai, Hidekazu Katayama, and Michio Iwaoka, Tokai University) by application of the native chain assembly method [15]. H89, a protein kinase A (PKA) inhibitor (vehicle; DMSO) was purchased from Tocris Bioscience (Bristol, UK).

### 2.2. Immunohistochemistry

Paraffin-embedded tissues were cut into 5-µm-thick sections and mounted on slides. The immunostaining of endometriotic lesions was performed in formalin-fixed, paraffin-embedded sections using specific antibodies to LGR-7 (1:100 dilution, LSBio Co., Seattle, WA, USA). Antigen retrieval was performed in 10 mM sodium citrate buffer (pH 6.0) by microwaving for 10 min and cooling to room temperature. The sections were stained with antibody or control IgG as a negative control using an Envision+ System/HRP rabbit (DAB+) kit (Dako Co, Tokyo, Japan). Slide staining with the first and second antibodies was performed according to the manufacturer’s instructions.

### 2.3. Isolation and Culture of Human Endometriotic Stromal Cells (ESCs)

Primary ESC culture was conducted as described [16]. Briefly, endometriotic tissue was dissected free of underlying parenchyma, minced into small pieces, incubated in DMEM F12 with type I collagenase (2.5 mg/mL) and DNase I (15 U/mL) for 1–2 h at 37 °C, and separated using serial filtration. Debris was removed using a 100 nm nylon cell strainer (Becton Dickinson, Lincoln Park, NJ, USA), and dispersed epithelial glands were eliminated with a 70 µm nylon cell strainer. Stromal cells remaining in the filtrate were collected by centrifugation, resuspended in DMEM/F12 with 10% charcoal-stripped FBS, penicillin (100 U/mL), streptomycin (100 ng/mL), and amphotericin B (250 ng/mL), plated onto 100 mm dishes (Iwaki Co, Chiba, Japan). When the cells became confluent after two days, they were dissociated with 0.25% trypsin, harvested by centrifugation, and replanted in six-well plates at 2 × 10^5^ cells/well. They were kept at 37 °C in a humidified 5% CO_2_/95% air environment until they were grown to confluence. Purification of the stromal cell population was confirmed by immunocytochemical staining for the following antibodies: vimentin (stromal cells), cytokeratin (epithelial cells), and CD45 (monocytes and other leukocytes). The purity of the stromal cell was more than 98%, as judged by positive cellular staining for vimentin and negative cellular staining for cytokeratin and CD45.

### 2.4. Reverse Transcription (RT), Conventional Polymerase Chain Reaction (PCR), and Quantitative Real-Time Polymerase Chain Reaction (PCR) Analysis

Total RNA was extracted from human ESCs using the ISOGEN-II (NIPPON GENE, Tokyo, Japan). RT was performed using Rever Tra Ace qPCR RT Master Mix with gDNA Remover (TOYOBO, Tokyo, Japan). About 1.0 μg of total RNA was reverse-transcribed in a 20-μL volume. For the quantification of various mRNA levels, conventional PCR was performed with a thermal cycler. PCR cycle number is 25 and 35 for glyceraldehyde 3-phosphate dehydrogenase (GAPDH) and LGR-7, respectively. Real-time PCR was performed using the Mx3000P Real-Time PCR System (Agilent Technologies, Santa Clara, CA, USA) according to the manufacturer’s instructions [17]. The PCR primers were selected from different exons of the corresponding genes to discriminate PCR products that might arise from possible chromosomal DNA contaminants. The expression of each mRNA was normalized by GAPDH mRNA. The following PCR primers were used: LGR-7 primer (NM_021634.4; 1851-1880 and 2078-2051), GAPDH primer (NM_002046-7; 602-621 and 1053-1034), collagen-I primer (NM_001353802-1; 501-520 and 640-621), PAI-1 primer (NM_002438-4; 307-287 and 212-229), α-SMA primer (NM_002438-4; 307-287 and 212-229), and CTGF primer (NM_000576-2; 52-71 and 319-300). Amplification was performed with 30 cycles of denaturing (98 °C, 10 s), annealing (60 °C, 10 s), and extension (72 °C, 10 s).

### 2.5. Measurement of Cyclic AMP (cAMP) and IL-8

The concentration of cAMP in conditioned media was measured using a cAMP immunoassay kit (R&D, Minneapolis, MN, USA), and that of IL-8 was measured using a specific ELISA kit (R&D). The sensitivities of the assays were 1.5 pmol/mL and 7.5 pg/mL for cAMP and IL-8, respectively. The intra-assay and inter-assay coefficients of variation were less than 5% in these assays.

### 2.6. Western Blotting

Cultured cells were homogenized in the lysis buffer containing 50 mM Tris HCl (pH 6.8), 2% sodium dodecyl sulfate, 10% glycerol, 50 mM dithiothreitol, and 0.1% bromophenol blue and diluted to 1 mg total protein/mL. Samples were resolved by 10% SDS-PAGE. Proteins were blotted onto a nitrocellulose membrane and incubated with rabbit antibody to total human p38 MAPK or rabbit antibody to phospho-specific (Thr180/Tyr182) p38MAPK (1:1000; New England Biolabs, Inc., Beverly, MA, USA) as a primary antibody, and anti-rabbit antibody (1:1000 Santa Cruz Biotechnology, Inc., Santa Cruz, CA, USA) as a secondary antibody. Phosphorylation of Thr180/Tyr182 is known to activate p38 MAPK [18]. Immune complexes were visualized by the use of an ECL Western blotting system (Amersham Pharmacia Biotech, Little Chalfont, UK).

### 2.7. Collagen Gel Contraction Assay

Cellular collagen gel contraction assays were performed as previously described [19]. A sterile solution of acid-soluble collagen type I purified from porcine tendons (Cellmatrix type I-A; NittaGelatin Inc., Osaka, Japan) was prepared according to the manufacturer’s instructions. ESCs were embedded in collagen gel and cultured three-dimension. Briefly, ESCs were suspended in the collagen solution (5.0 × 10^5^ cells/mL). The collagen/cell mixture (2 mL/plate) was dispensed into 35-mm culture plates (Corning, New York, NY, USA) coated with 0.2% bovine serum albumin (BSA) and the mixture was allowed to polymerize at 37 °C for 30 min. After polymerization, 1 mL of culture medium containing 10% FBS with or without RLX-2 (100 ng/mL) was added to each plate. In some experiments, H-89, a PKA inhibitor (5 μM), was added to the culture media. After incubation for 24 h, the collagen gels were photographed, and the gel surface area was measured with the public domain Image program 1.61 developed at the National Institutes of Health (Bethesda, MD, USA). In some experiment, after 24 h of RLX-2 treatment, culture medium was replenished with control culture medium (Figure 3c).

### 2.8. Endometrial Transplantation Mouse Model

An endometriosis mouse model using endometrial transplantation was performed as previously described (Figure 5a) [20,21,22]. Donor and recipient mice were bilaterally ovariectomized through paravertebral incisions to exclude endogenous estrogen and menstrual cycle. All donor and recipient mice received subcutaneous (s.c.) injections of estradiol dipropionate (100 mg/kg) in sesame oil (Obahormone depot; Aska, Tokyo) every week from the time of ovariectomy. Seven days after the ovariectomy, the uterine horns from the donor were removed, trimmed of connective tissue, and opened longitudinally in a tissue culture dish containing DMEM/F-10 at 37 °C, supplemented with 100 U/mL penicillin and 100 mg/mL streptomycin. Four round endometrial fragments (3 mm in diameter), including the myometrium, were collected using a biopsy punch (Kai Medical, Tokyo, Japan). The endometrial tissues were transplanted to the peritoneal wall of recipient mice with a 7-0 polypropylene suture (Ethicon, Johnson & Johnson, Tokyo, Japan). This location was chosen because it is in contact with the endometrial surface epithelium of the implants and peritoneum [22]. The wound was closed with a 3-0 suture, and mice were placed on a warming carpet to prevent hypothermia. The day of implantation was defined as Day 0, and from day 7, we sacrificed a few mice to confirm the formation of endometriotic-like lesions. Then the recipient mice were treated with RLX-2 (1 µg/g/day) or 0.1% BSA/PBS (control) every day for seven days. The mice were euthanized under anesthesia on Days 14 post-implantation. The endometrial implants were removed and captured by digital photographs to measure the lesion area. Evaluation of fibrosis at endometriotic-like lesions was performed by Masson Trichrome staining [23].

### 2.9. Statistical Analysis

Data were evaluated by Mann-Whitney using JMP version 10. *p* value less than 0.05 was accepted as statistically significant.

## 3. Results

### 3.1. Expression of LGR-7 in Endometriotic Lesions

To evaluate the expression and the localization of LGR-7, a primary receptor of RLX-2 in ESCs and endometriotic lesions, conventional PCR and immunohistochemistry were performed. LGR-7 mRNA was expressed in ESCs obtained from all three patients, although the expression levels were different among individuals (Figure 1a). LGR-7 protein was expressed in both epithelial and stromal cells in endometriotic lesions (Figure 1b).

### 3.2. The Effect of RLX-2 for Endometriotic Stromal Cells (ESCs)

We confirmed a dose-dependent increase of cAMP concentration in the supernatant of ESCs stimulated with RLX-2 (Figure 2a). Because the endometriotic lesions grow with fibrosis, we examined mRNA expression of several fibrotic related factors such as collagen-I [24,25], plasminogen activator inhibitor-I(PAI-1) [26,27], connective tissue growth factor [28] (CTGF), and α-smooth muscle actin (α-SMA) [28] by quantitative-PCR in RLX-2-treated cells. We found that the mRNA expressions of collagen-I and PAI-1 were decreased to around 50–60% in RLX-2-treated group compared to the control (*p* < 0.05), while those of CTGF and α-SMA were not changed (Figure 2b). We also checked the inflammatory cytokine IL-8, one of the elevated cytokines in endometriosis patients [29]. IL-8 protein concentration in cultured supernatant was significantly lower in 24 h-RLX-2 treated group compared to that in the control (*p* < 0.05, Figure 2c).

### 3.3. The Effect of RLX-2 for Collagen Gel Contractility of ESCs

As collagen gel contractility, a model of scar formation, is enhanced in human ESCs [19], we used this assay model to assess the effect of RLX-2. As shown in Figure 3a, a shrink of the collagen gel was confirmed in the control group, which is canceled by RLX-2 treatment. As shown in Figure 3b, the area of the gel of RLX-2 treated group was significantly larger compared to the control. H89, a protein kinase A (PKA) inhibitor, cancelled the suppressive effect of RLX-2 on gel contraction (Figure 3a,b). The inhibitory effect of RLX-2 on the contraction was eliminated by removing RLX-2 from the medium (Figure 3c), suggesting that ESCs embedded in collagen gel were viable after RLX-2 treatment. We next examined the phosphorylation of p38 MAPK in ESCs by Western blotting. A suppression in p38 MAPK phosphorylation was found in RLX-stimulated ESCs, although the amount of total p38 MAPK was kept unchanged for 24 h (Figure 4).

### 3.4. Endometriosis Mouse Model (Endometrial Implantation Mouse Model)

In the endometriosis mouse model (Figure 5a), administration of RLX-2 significantly decreased the area of the endometriotic-like lesion compared to the non-treated control (*p* = 0.01, Figure 5b). As RLX-2 has an anti-fibrotic potential [11], the extent of fibrosis in the mouse endometriotic-like lesions was evaluated using the Masson trichrome staining method. As shown in Figure 5c, fibrous tissues, which are stained with light blue, were decreased in the RLX-2 group compared to the control.

## 4. Discussion

Endometriosis, an estrogen-dependent disease, improves during pregnancy, and high progesterone milieu during pregnancy may suppress endometriosis [30]. However, it is also known that the expression levels of progesterone receptor (PR)-B, which is an active form of PR, are low [31,32,33] and sometimes even undetectable [34] in endometriotic lesions. The mechanisms of PR deficiency are multiple including inflammatory condition, epigenetic and miRNAs [1]. PR deficiency causes the development of progesterone resistance in women who no longer respond to progesterone [35]. Therefore, during pregnancy in addition to progesterone effect, other factor(s) might be involved in the improvement of endometriosis.

RLX-2 is highly expressed during pregnancy. Therefore, in the present study, we focused our attention on RLX-2 for the treatment of endometriosis. The primary RLX-2 receptor identified was the leucine-rich G protein-coupled receptors 7, later renamed RLX-2 family peptide receptor (RXFP) 1 [36]. Upon activation of LGR-7, increased cAMP production and activation of cAMP-dependent protein kinase (PKA) are well known [37].

Morelli et al. reported that all endometriotic tissues express LGR-7 mRNA, and whose expression levels are lower than eutopic endometrium [14]. Nevertheless, the significance of LGR-7 expression in endometriosis has remained to be tested. In the present study, we confirmed the presence of LGR-7 at mRNA and protein levels in endometriotic lesions, and cAMP production in endometriotic cells in vitro with recombinant RLX-2 stimulation suggesting that RLX-2 was effective on ESCs. Then, we have investigated the various effect of RLX-2 on endometriotic cells, mainly focusing on fibrosis (collagen-I and PAI-1) and inflammation (IL-8). Type I collagen is reported to be a significant contributor to endometriosis-associated fibrosis [25,38] and could be a good target for the treatment in endometriosis. PAI-1 plays a vital role in coagulation/fibrinolysis and fibrosis. High concentrations of PAI-1 in the intraperitoneal fluid of patients with endometriosis contribute to the development of peritoneal lesions [26,27]. IL-8, a key player in endometriosis, is increased in peritoneal fluid of endometriosis patients and stimulates proliferation, matrix metalloproteinase activity, invasive capability, anti-apoptotic effect, and adhesion capability of endometrial stromal cells [29,39,40]. We found that RLX-2 suppressed collagen-I, PAI-1, and IL-8 expression in ESCs.

As a scar formation model, we have utilized the gel contraction assay where fibroblasts generate the contractile forces propagate throughout the collagen matrix and arrange collagen fibers to higher density structure, resulting in decreased matrix volume [41]. Using this assay, Yuge et al. proved that ESCs possess strong contractile to form a scar in endometriosis [19]. In the present study, the contractile forces of ESCs were suppressed with RLX-2 treatment significantly, which was reversible by depletion of the ligand or by adding H89, a PKA inhibitor, suggesting that RLX-2 had no cell-toxicity effect, and activated PKA, respectively. This reversible effect of RLX-2 may also be associated with the relapse of endometriosis symptom after some time of partum.

Our group has previously advocated that the activation of p38 MAPK is critical in the pathophysiology of endometriosis by upregulating IL-8 production and increasing cell proliferation in ESCs [16]. Inconsistent with the notion, we and other groups confirmed p38 MAPK inhibitors could be a suppressor of endometriosis in mice model [42,43]. Nevertheless, it still takes time to use p38 MAPK inhibitors in clinical practice [44]. An alternative to direct suppression of p38 MAPK by its inhibitors, we had found PKA activation through decidualization of endometrium could be a suppressor of p38 MAPK [45]. In the present study, we have confirmed that RLX-2 increased the production of cAMP and decreased the phosphorylation of p38MAPK in ESCs. In some experiments, H89, an inhibitor of PKA cancelled the RLX-2 effect. Collectively, RLX-2 might suppress endometriosis related events by suppressing p38 MAPK via cAMP/PKA activation in ESCs.

In the present study, we confirmed the effect of RLX-2 on in vivo mouse model of endometriosis. RLX-2 decreased the size of endometriotic-like lesions. As a reason for diminishing the size of lesions with RLX-2 stimuli, the fibrotic component in stromal lesions was decreased, confirmed by Masson trichrome staining method.

In the series of present data, we found that RLX-2 inhibited fibrosis, scar formation, and inflammation in endometriosis. But we have not been able to examine the efficacy of RLX-2 on pre-existing fibrosis. RLX-2 is expected to use in various fibrous diseases [11], and is also known to enhance heart function with dilating peripheral blood vessels, and clinical research was held as a therapeutic agent for heart failure [46]. Although further study is needed, RLX-2 can be expected to be a new therapy for endometriosis with various effects.

## Figures and Tables

**Figure 1 biomedicines-08-00467-f001:**
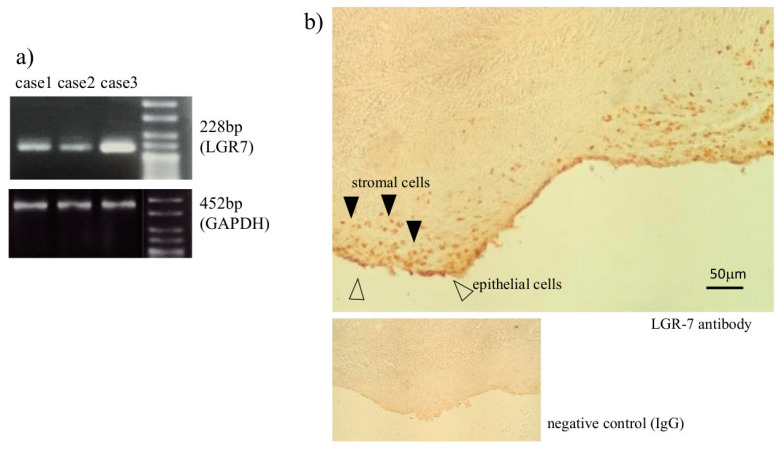
The expression of LGR-7 in the endometriotic lesion. Endometriotic stromal cells (ESCs) were obtained from three endometriomas and analyzed by conventional PCR for the mRNA expression of LGR-7, a receptor of relaxin (RLX)-2, and GAPDH (**a**). The expression of LGR-7 protein detected by immunohistochemistry in endometriomas is shown. Arrowheads indicated the expression of LGR-7 in stromal cells (▲) and epithelial cells (△). Rabbit IgG was used as a negative control (**b**).

**Figure 2 biomedicines-08-00467-f002:**
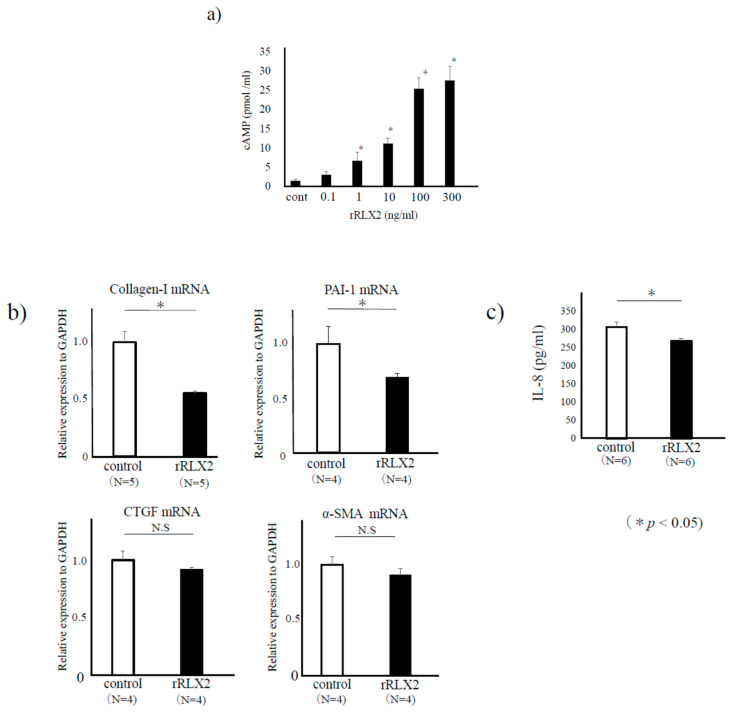
The effect of relaxin (RLX)-2 on endometriotic stromal cells (ESCs). ESCs were incubated with RLX-2 for 24 h, and the concentration of cAMP in conditioned media was measured by a specific assay kit. Representative data from three different experiments were shown as the mean ± SEM *: *p* < 0.05 compared to the control (**a**). After 8 h stimulation with recombinant-RLX-2 (rRLX-2, 100 ng/mL), real-time RT-PCR was performed to measure the mRNA expression levels of collagen-I, plasminogen activator inhibitor-1 (PAI-1), connective tissue growth factor (CTGF), and α-smooth muscle actin (α-SMA) in the control and RLX-2-treated group. Data were normalized by GAPDH mRNA levels to show the relative abundance. Representative data from three different experiments were shown as the mean ± SEM relative to an adjusted value of 1.0 for the mean value of the control (**b**). ESCs were incubated with RLX-2 for 24 h, and the concentration of IL-8 (pg/mL) in conditioned media was measured by specific assay kits (**c**). *: *p* < 0.05.

**Figure 3 biomedicines-08-00467-f003:**
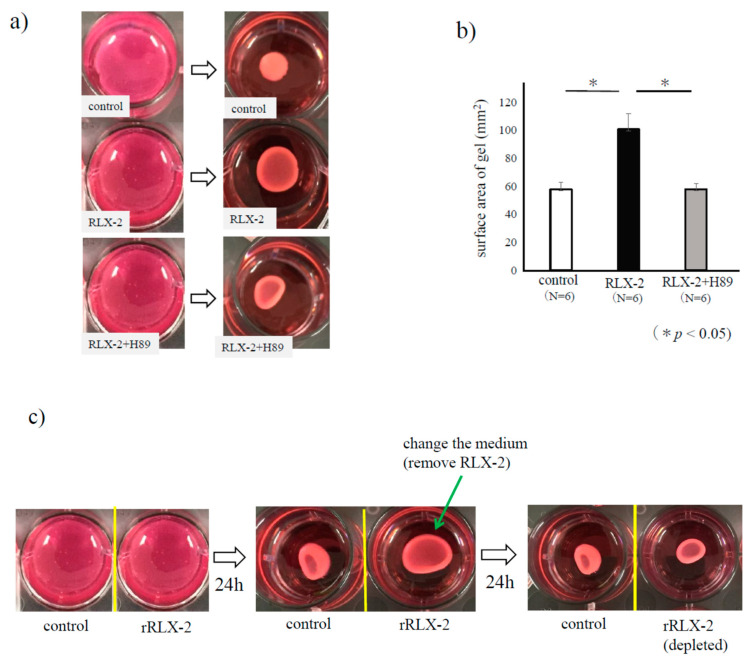
The relaxin (RLX)-2 effect on collagen gel contractility of endometriotic stromal cells (ESCs). Collagen gel contraction assay was performed to evaluate the effect of RLX-2 on the contractility of ESCs in the presence of 10% FBS. After 24 h-s RLX-2 stimulation for ESCs, gel areas were approximated by multiplying the major axis diameter by the minor axis diameter in the control and RLX-2 group. In some experiments, ESCs were treated with 5 μM of H89, an inhibitor of Protein Kinase-A, for 1 h, followed by the treatment of RLX-2 (100 ng/mL) for 24 h (**a**). Data from three different experiments were shown as the mean ± SEM relative *: *p* < 0.05 (**b**). To check the viability of ESCs after RLX-2 stimulation of 24 h, RLX-2 was removed from medium, and ESCs were cultured with control medium for another 24 h (**c**).

**Figure 4 biomedicines-08-00467-f004:**
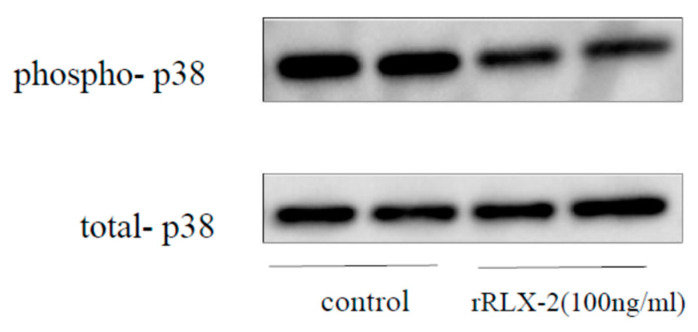
p38MAPK phosphorylation with relaxin-2 (RLX-2) treatment in endometriotic stromal cells (ESCs). ESCs were incubated with RLX-2 for 24 h. Cell lysates were prepared and assayed for phosphorylated p38 MAPK (phospho-p38) or total p38 MAPK (total-p38) by Western blotting. The result is representative of three separate experiments.

**Figure 5 biomedicines-08-00467-f005:**
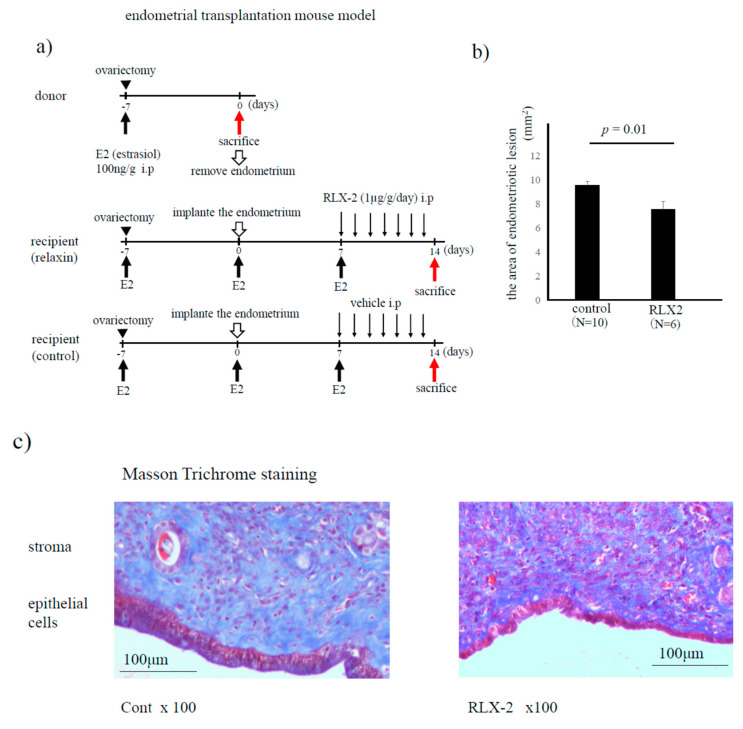
Endometrial transplantation mouse model. Both donor and recipient mice were treated with estradiol (E2) once a week from the day of ovariectomy. Seven days after the ovariectomy, endometrial fragments from donor mice were transplanted into the peritoneum of recipient mice. Seven days after the transplantation, we sacrificed a few mice to confirm the formation of endometriotic lesions. Then, the recipient mice were treated with RLX-2 (1 µg/g/day) or 0.1% BSA/PBS (control) every day. Fourteen days after transplantation, we sacrificed recipient mice, and tissue samples were used following experiments (**a**). The area of collected endometriotic-like lesions (control group *N* = 10, and RLX-2 group *N* = 6) was compared (**b**), and the extent of fibrosis (stained with blue color) was evaluated by the Masson trichrome staining method in control and RLX-2 groups (**c**).

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
