# Peer review of "Relaxin-2 May Suppress Endometriosis by Reducing Fibrosis, Scar Formation, and Inflammation"

_biomedicines, 2020, doi:10.3390/biomedicines8110467_

Round 1

Reviewer 1 Report

I read with great interest the manuscript, which falls within the aim of this Journal. In my honest opinion, the topic is interesting enough to attract the readers’ attention. Nevertheless, authors should clarify some points and improve the discussion, as suggested below.

Authors should consider the following recommendations:

  • Manuscript should be further revised in order to correct some typos and improve style.
  • Accumulating evidence suggests that immune cells, adhesion molecules, extracellular matrix metalloproteinase and pro-inflammatory cytokines activate/alter peritoneal microenvironment, creating the conditions for differentiation, adhesion, proliferation and survival of ectopic endometrial cells. I would discuss these points in the light of new theories about the pathogenesis of endometriosis, referring to: PMID: 31717614; PMID: 31663401.

Author Response

Manuscript should be further revised in order to correct some typos and improve style.

>Thank you very much for your favorable comments and suggestion. In the revised version, we have corrected typos and edited the manuscript.  Please find the changes which are underlined in the attached revised paper. 

Accumulating evidence suggests that immune cells, adhesion molecules, extracellular matrix metalloproteinase and pro-inflammatory cytokines activate/alter peritoneal microenvironment, creating the conditions for differentiation, adhesion, proliferation and survival of ectopic endometrial cells. I would discuss these points in the light of new theories about the pathogenesis of endometriosis, referring to: PMID: 31717614; PMID: 31663401.

>Thank you very much for your suggestion, and we found that the papers you suggested are very informative and added value to our submitting paper.  

In the revised paper, we added the sentences below. 

(line 76, introduction part)

Accumulating evidence suggests that immune cells, adhesion molecules, extracellular matrix metalloproteinase and pro-inflammatory cytokines activate or alter peritoneal microenvironment, creating the conditions for the development of ectopic endometrial cells1. (PMID: 31717614)

 (line87,introduction part)

Interestingly, Lagana et al. reported that by analyzing macrophages, pro-inflammatory macrophages are dominant in the early stages of endometriosis, while pro-fibrotic macrophages are activated in the late stages6. PMID: 31663401

(line 384, discussion part)

The mechanisms of PR deficiency are multiple including inflammatory condition, epigenetic and miRNAs1.(PMID: 31717614)

Reviewer 2 Report

I had the honor to review this manuscript. I found the manuscript interesting, well done, and informative. The authors acknowledge that further investigation is needed before considering the use of relaxing in clinical practice. However, I believe that this study is leading research in endometriosis in the right direction. I am happy to recommend acceptance.

Author Response

I had the honor to review this manuscript. I found the manuscript interesting, well done, and informative. The authors acknowledge that further investigation is needed before considering the use of relaxing in clinical practice. However, I believe that this study is leading research in endometriosis in the right direction. I am happy to recommend acceptance.

> Thank you very much for your favorable comment.  In the revised version, we have corrected typos and edited the manuscript. We believe the paper is improved.